# Oral supplementation of gut microbial metabolite indole-3-acetate alleviates diet-induced steatosis and inflammation in mice

Yufang Ding[1†], Karin Yanagi[2†], Fang Yang[1], Evelyn Callaway[1], Clint Cheng[1], Martha E Hensel[3], Rani Menon[1], Robert C Alaniz[4], Kyongbum Lee[2]*, Arul Jayaraman[1,4]*

[1]Artie McFerrin Department of Chemical Engineering, Texas A&M University, College Station, United States; [2]Department of Chemical and Biological Engineering, Tufts University, Medford, United States; [3]Department of Pathobiology, College of Veterinary Medicine and Biomedical Sciences, Texas A&M University, College Station, United States; [4]Department of Microbial Pathogenesis and Immunology, College of Medicine, Texas Health Science Center, Texas A&M University, Bryan, United States

*For correspondence:
kyongbum.lee@tufts.edu (KL);
arulj@mail.che.tamu.edu (AJ)

†These authors contributed equally to this work

Competing interest: The authors declare that no competing interests exist.

**Abstract** Non-alcoholic fatty liver disease (NAFLD) is the most common chronic liver disease in Western countries. There is growing evidence that dysbiosis of the intestinal microbiota and disruption of microbiota-host interactions contribute to the pathology of NAFLD. We previously demonstrated that gut microbiota-derived tryptophan metabolite indole-3-acetate (I3A) was decreased in both cecum and liver of high-fat diet-fed mice and attenuated the expression of inflammatory cytokines in macrophages and *Tnfa* and fatty acid-induced inflammatory responses in an aryl-hydrocarbon receptor (AhR)-dependent manner in hepatocytes. In this study, we investigated the effect of orally administered I3A in a mouse model of diet-induced NAFLD. Western diet (WD)-fed mice given sugar water (SW) with I3A showed dramatically decreased serum ALT, hepatic triglycerides (TG), liver steatosis, hepatocyte ballooning, lobular inflammation, and hepatic production of inflammatory cytokines, compared to WD-fed mice given only SW. Metagenomic analysis show that I3A administration did not significantly modify the intestinal microbiome, suggesting that I3A's beneficial effects likely reflect the metabolite's direct actions on the liver. Administration of I3A partially reversed WD-induced alterations of liver metabolome and proteome, notably, decreasing expression of several enzymes in hepatic lipogenesis and β-oxidation. Mechanistically, we also show that AMP-activated protein kinase (AMPK) mediates the anti-inflammatory effects of I3A in macrophages. The potency of I3A in alleviating liver steatosis and inflammation clearly demonstrates its potential as a therapeutic modality for preventing the progression of steatosis to non-alcoholic steatohepatitis (NASH).

## eLife assessment

The studies are **important** to the field of hepatic steatosis and inflammation. The data provided are **convincing** that treatment with I3A mitigated Western diet (WD)-induced hepatic steatosis, inflammation and reversed WD-induced alterations in liver bile acids and free fatty acids in mice.

## Introduction

Non-alcoholic fatty liver disease (NAFLD) is the most common chronic liver disease in Western countries, with a prevalence rate of 21–25% in North American and Europe (*Loomba and Sanyal, 2013*). It is a multi-stage disease (*Day, 2011*) that can progress from liver steatosis (characterized by macrovesicular fat deposition), which is benign and reversible (*Day, 2011*; *Tiniakos et al., 2010*), to more severe forms of the disease such as non-alcoholic steatohepatitis (NASH) and fibrosis. Approximately 25% of individuals having liver steatosis develop NASH, which is characterized by liver inflammation, dysregulated lipid metabolism, and cell damage (*Tiniakos et al., 2010*). A subset of NASH patients develops cirrhosis and even hepatocellular carcinoma (*Calzadilla Bertot and Adams, 2016*). Although significant progress has been made in understanding the pathogenesis of NAFLD, factors leading to the progression from steatosis to NASH remain poorly understood, and there are currently no pharmacological treatments available for NASH.

There is growing evidence that dysbiosis of the intestinal microbiota and disruption of microbiota-host interactions contribute to the pathology of NASH (*Brandl and Schnabl, 2017*; *Boursier et al., 2016*; *Loomba et al., 2017*; *Aron-Wisnewsky et al., 2020*). Certain shifts in the intestinal microbiota community composition, e.g., expansion of the phyla Verrucomicrobia and Proteobacteria, correlate with NASH in both human and animal studies (*Hoyles et al., 2018*; *Le Roy et al., 2013*). One potential mechanism linking intestinal microbiota dysbiosis and NASH is compromised intestinal barrier integrity, which promotes the translocation of bacterial products from the lumen to circulation. This can contribute directly and indirectly (through intestinal inflammation) to liver inflammation (*Saltzman et al., 2018*; *Bibbò et al., 2018*; *Ding et al., 2019*). Additionally, microbial dysbiosis alters the balance of bioactive metabolites produced by gut bacteria such as bile acids, short chain fatty acids, and aromatic amino acid derivatives, which have been shown to impact liver metabolism and inflammation in NAFLD through host receptor-mediated pathways (*Ding et al., 2019*).

In our previous study (*Krishnan et al., 2018*), we demonstrated that gut microbiota-derived tryptophan metabolites indole-3-acetate (I3A) and tryptamine (TA) were decreased in both cecum and liver of high-fat diet (HFD)-fed mice compared to low-fat diet (LFD)-fed control mice (CN). In vitro, both I3A and TA attenuated the expression of inflammatory cytokines (*Tnfa*, *Il1b*, and *Ccl2*) in macrophages exposed to palmitate and LPS. In hepatocytes, I3A significantly attenuated *Tnfa* and fatty acid-induced inflammatory responses in an aryl-hydrocarbon receptor (AhR)-dependent manner. Based on these findings, we hypothesized that I3A could protect against NAFLD progression in vivo. We show that supplementation of I3A in drinking water alleviates liver steatosis and inflammation even when mice are continued on the NAFLD-inducing diet, and that these effects correlate with a decrease in both lipogenesis and β-oxidation in the liver. We also demonstrate that the anti-inflammatory effects of I3A in macrophages can be mediated by AMP-activated protein kinase (AMPK).

## Results

### Oral administration of I3A alleviates diet-induced hepatic steatosis and inflammation

We utilized a mouse model of diet-induced fatty liver disease (*Asgharpour et al., 2016*) to investigate the effect of I3A administration through drinking water (*Figure 1—figure supplement 1A*) on liver steatosis and inflammation. Similar to our previous finding in HFD-fed mice (*Krishnan et al., 2018*), the levels of I3A were significantly reduced in fecal material (50% and 64% decrease at weeks 8 and 16, respectively) and the liver (70% decrease at week 16) of Western diet (WD)-fed mice compared to CN fed an LFD (*Figure 1A and B*, *Figure 1—figure supplement 1C*). Mice that were given WD and sugar water (SW) containing I3A showed increased I3A levels in fecal material, the liver and serum compared to mice given the same WD and SW without I3A (*Figure 1A, B, and C*). Next, we sought to determine if the I3A administration impacted WD-induced features of NAFLD. Compared to CN, WD-fed mice had elevated serum alanine aminotransferase (ALT) levels, indicating liver injury. Treatment of WD-fed mice with the high dose of I3A (WD-100) significantly reduced serum ALT levels 2 weeks after beginning the I3A treatment (week 10, *Figure 1D*) and beyond. Additionally, I3A treatment reduced liver triglycerides (TG) in a dose-dependent manner at week 16 (*Figure 1E*). Scoring of hematoxylin and eosin (H&E)-stained liver sections indicated that the I3A treatment improved liver steatosis, hepatocyte ballooning, and lobular inflammation in a dose-dependent manner (*Figure 1F*

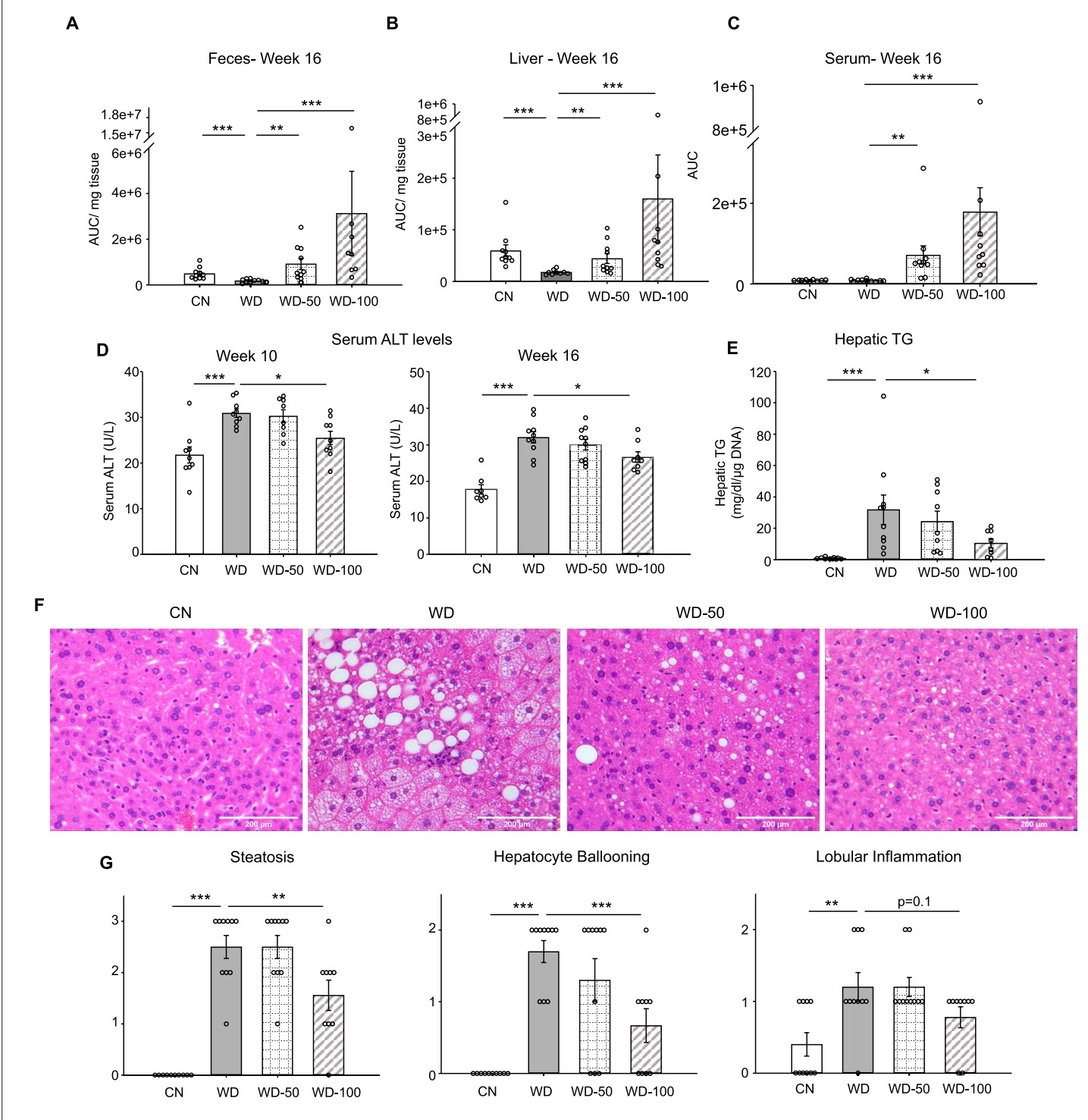

**Figure 1.** Oral administration of indole-3-acetate (I3A) alleviates diet-induced hepatic steatosis and inflammation. (**A**) Fecal, (**B**) liver, and (**C**) serum concentrations of I3A in male B6 129SF1/J mice in control low-fat diet (CN), Western diet (WD), WD with low-dose I3A (WD-50), and WD with high-dose I3A (WD-100) at week 16. (**D**) Serum alanine aminotransferase (ALT) levels in mice at weeks 10 and 16. (**E**) Liver triglyceride (TG) levels at week 16. Data shown are TG concentrations (mg/dl) normalized to corresponding tissue DNA contents (μg DNA). (**F**) Representative liver sections stained with hematoxylin-eosin (H&E). (**G**) Histology score for steatosis, hepatocyte ballooning, and lobular inflammation. H&E-stained liver sections were evaluated by an expert pathologist using the non-alcoholic steatohepatitis (NASH) CRN and fatty liver inhibition of progression (FLIP) consortia criteria. Data shown are mean ± SEM (n=10 per group). *: p<0.05, **: p<0.01, ***: p<0.001 using Wilcoxon rank sum test.

The online version of this article includes the following figure supplement(s) for figure 1:

*Figure 1 continued on next page*

*Figure 1 continued*

**Figure supplement 1.** Overview of in vivo study.

*and G*). Treatment with I3A didn't alter weight increase nor food intake compared to WD-fed mice (*Figure 1—figure supplement 1D*).

We also investigated whether I3A treatment modulated the levels of inflammatory cytokines in WD-fed mice. All pro-inflammatory cytokines in the panel (e.g. *Tnfa*, *Il-6*, *Ccl2*) were significantly elevated in livers of WD mice compared to CN (*Figure 2A* and *Figure 2—figure supplement 1A*). Treatment with I3A reduced the expression of these cytokines in a dose-dependent manner. Interestingly, WD upregulated IL-10, considered an anti-inflammatory cytokine, and I3A treatment downregulated this cytokine (*Figure 2—figure supplement 1A*). Taken together, these results demonstrate that I3A, provided via drinking water, attenuates diet-induced hepatic steatosis, cellular injury, and inflammation in mice.

## I3A reverses diet-induced alterations in liver bile acids and free fatty acids

Studies in mice and humans showed that NAFLD is associated with alterations in liver bile acids. Previously, we found that the ratio of chenodeoxycholic acid (CDCA) to cholic acid (CD) greatly decreased when HepG12 and AML12 cells were treated with I3A, indicating that I3A can directly alter host cell bile acid metabolism in vitro (*Krishnan et al., 2018*). In the present study, mice in the WD group showed reduced liver total bile acids and CDCA-derived bile acids compared to CN (*Figure 2B*). Treatment with I3A led to a further decrease in the CDCA-derived bile acids, consistent with our previous observation of I3A's effect on hepatocytes in vitro. We also measured the concentrations of nine major free fatty acids (FFAs; Lauric acid, Myristic acid, Palmitic acid, Palmitoleic acid, Stearic acid, Oleic acid, Linoleic acid, Linolenic acid and Arachidonic acid). In the liver, WD significantly increased the total concentration of these FFAs by 2.7-fold compared to CN (*Figure 2—figure supplement 1B*). Treatment with I3A had no significant effect, although the total FFA concentration trended lower in the WD-100 group (36% reduction, p=0.11, *Figure 2—figure supplement 1B*). The total FFA concentrations in serum were similar across all four groups (*Figure 2—figure supplement 1B*).

## I3A administration does not significantly alter the fecal microbial community as well as fecal metabolome

Next, we investigated whether the hepatoprotective effects of I3A were due to a modification of the gut microbiome. The fecal microbial communities at week 8 of WD (prior to I3A treatment) and week 16 (at termination) were analyzed using 16S rRNA sequencing. Fecal microbiome of WD-fed mice showed reduced α-diversity compared to CN, which was not further altered upon I3A treatment (*Figure 3—figure supplement 1A*). Similarity analysis using the Bray-Curtis dissimilarity metric showed that the microbiome compositions of CN and WD-fed mice were significantly different (*Figure 3—figure supplement 1B*). The fecal microbiome compositions of WD-fed mice did not change significantly upon I3A treatment (*Figure 3—figure supplement 1B*). Analysis of fecal microbiota taxonomic profiles at the phylum and genus level (*Figure 3—figure supplement 1C, D*) showed major shifts in bacterial abundance of fecal microbiota from WD-fed mice relative to CN. At week 16, the relative abundance of phylum Verrucomicrobia increased, whereas Bacteroidetes decreased. Linear discriminant analysis effect size (LEfSe) detected 18 taxa showing significant differences in relative abundance between the WD group and CN, including an expansion of the genus *Akkermansia* and reduction of family Muribaculaceae. In contrast, only three genera showed significant shifts in WD-fed mice upon I3A treatment (*Figure 3—figure supplement 1E*).

We also investigated whether administration of I3A altered the metabolic profile of the gut microbiota. Principal component analysis (PCA) of untargeted LC-MS data showed that the fecal metabolome of WD-fed mice was significantly different from CN, whereas the metabolomes of WD-100 and WD groups largely overlapped (*Figure 3—figure supplement 2A*). Although permutational multivariate analysis of variance (PERMANOVA) suggested a modest difference in the variance of metabolite levels between WD-50 and WD groups, Hotelling's $T^2$ test found that the means (centroids) were not significantly different. Clustering analysis using the k-means showed that fecal metabolites could be

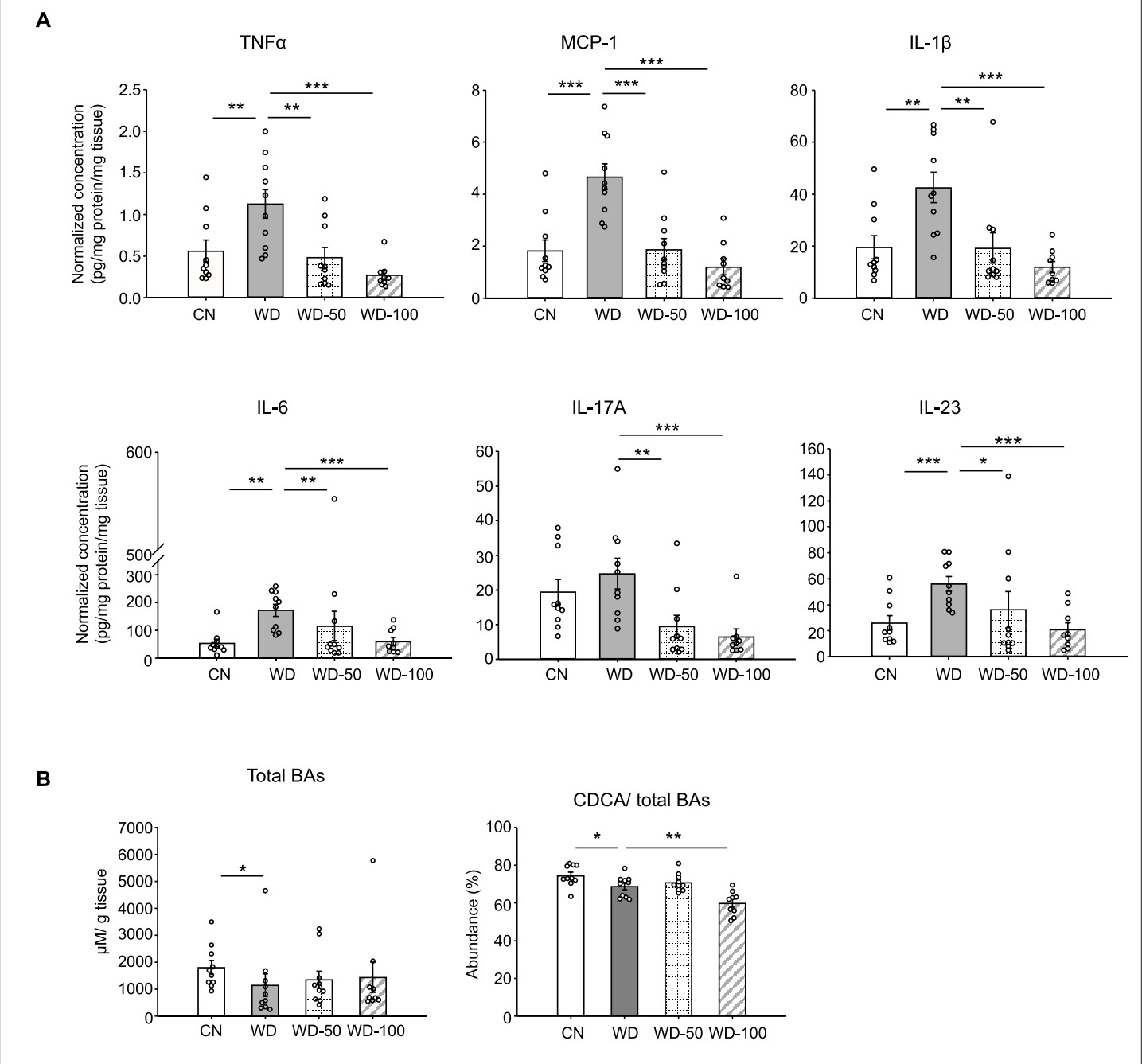

**Figure 2.** Indole-3-acetate (I3A) administration reverses Western diet (WD)-induced alterations in liver inflammatory cytokines and bile acids. (**A**) Inflammatory cytokines in liver tissue at week 16. (**B**) Liver total bile acid concentration (left panel), and abundance of chenodeoxycholic acid (CDCA) branch bile acids relative to total bile acids pool (right panel). Data shown are mean ± SEM. *: p<0.05, **: p<0.01, ***: p<0.001 using Wilcoxon rank sum test.

The online version of this article includes the following figure supplement(s) for figure 2:

**Figure supplement 1.** I3A supplementation reduces inflammatory cytokines and free fatty acid levels.

assigned into four groups (*Figure 3—figure supplement 2B*). We did not observe any clusters that showed a reversal of WD-induced changes in fecal metabolome by I3A treatment. Taken together, these results indicate that treatment with I3A had minimal effects on the fecal microbiota composition and metabolome of WD-fed mice.

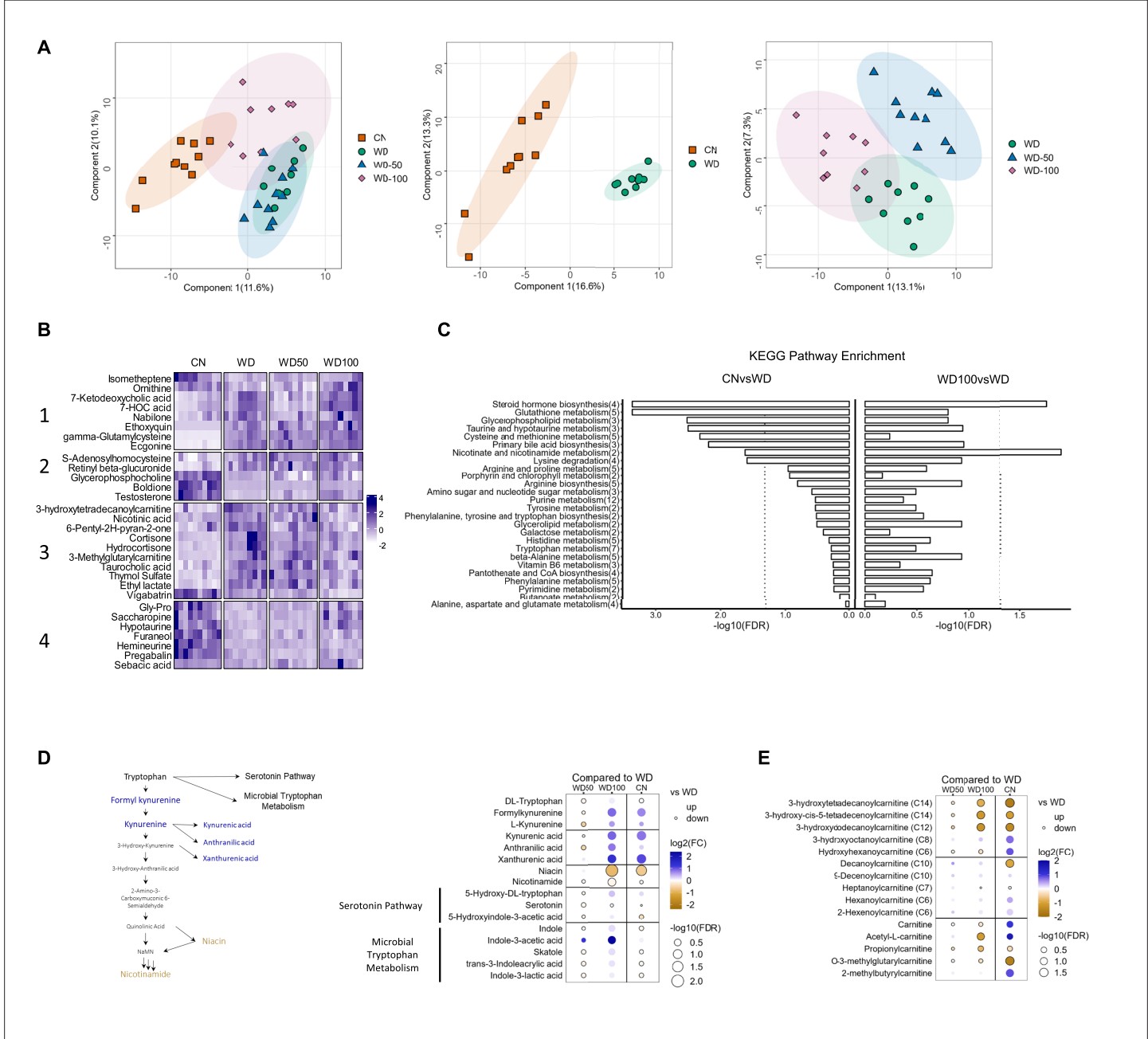

**Figure 3.** Indole-3-acetate (I3A) administration partially reverses diet-induced metabolome alterations in the liver. (**A**) Scatter plots of latent variable projections from PLS-DA of untargeted metabolomics data features. Comparison of all four experimental groups (left panel), control mice (CN) vs. Western diet (WD) group (middle panel), and WD vs. WD-50 and WD-100 groups (right panel). (**B**) Heatmap of significant metabolite features (FDR<0.1) based on statistical comparisons of treatment groups (CN vs. WD). (**C**) KEGG pathway enrichment analysis of the metabolites. Number in the parenthesis represents the metabolites detected in the pathway. (**D**) Schematic for tryptophan metabolism (left panel). Tryptophan metabolism metabolites fold-changes of WD-50, WD-100, CN relative to WD (right panel). (**E**) Acyl-carnitine fold-change of WD-50, WD-100, CN relative to WD. p-Values were calculated using Student's t-test and corrected by FDR.

The online version of this article includes the following figure supplement(s) for figure 3:

**Figure supplement 1.** 16srRNA metagenomics of mouse fecal microbiota.

**Figure supplement 2.** Untargeted metabolomic analysis of mouse fecal metabolites at week 16.

## High-dose I3A administration partially reverses diet-induced metabolome alterations in the liver

We next investigated the impact of I3A treatment on the liver metabolome using untargeted LC-MS. Projections of all annotated features onto latent variables from PLS-DA showed that the liver metabolome of WD-fed mice was significantly different from CN (*Figure 3A*). The low dose I3A (WD-50) group completely overlapped with the WD group, whereas the high dose (WD-100) group was significantly different from both WD and CN groups (*Figure 3A*). To further determine if the high dose of I3A was able to reverse the diet-induced alterations in liver metabolome, we performed a clustering analysis on all 30 annotated features that were significantly (FDR<0.1) different in WD mice compared to CN. This analysis placed the significant features into four groups. Group 1 comprised metabolites that were further increased or depleted by the high dose of I3A, whereas groups 3 and 4 comprised metabolites having WD-induced changes that were reversed by I3A treatment (except for vigabatrin) (*Figure 3B*). Enrichment analysis based on KEGG pathways (*Figure 3C*) showed that nicotinate and nicotinamide metabolism and steroid hormone biosynthesis were significantly altered by WD (CN vs WD), which were reversed by the high-dose I3A treatment (WD vs WD-100).

Because nicotinamide and nicotinate are products of tryptophan metabolism, we manually integrated the peak areas for all features we identified as tryptophan metabolites (*Figure 3D*). This added formylkynurenine, kynurenic acid, and xanthurenic acid to the list of annotated metabolites comprising the kynurenine branch of tryptophan metabolism. The high dose of I3A restored WD-induced depletion of metabolites in the upper kynurenine branch, and ameliorated WD-induced increases in niacin and nicotinamide in the lower branch. These effects were dose-dependent, as they were not present in the WD-50 group.

Acyl-carnitines comprise another class of metabolites having abundances that were returned to CN levels in WD-fed mice by the high dose of I3A (*Figure 3E*). Compared to CN, mice in the WD group showed elevated levels of long-chain 3-hydroxyacyl-carnitines. These acyl-carnitines are closely linked to 3-hydroxyacyl-CoAs, which are intermediates of beta-oxidation, and their accumulations suggest incomplete beta-oxidation (*Su et al., 2005*; *Natarajan and Ibdah, 2018*). Treatment with the high dose of I3A significantly reversed these accumulations. As was the case for tryptophan metabolites, the effect of I3A on acyl-carnitines was dose-dependent.

## I3A administration partially reverses diet-induced proteome alterations in the liver

We next investigated if I3A also changed the expression levels of liver proteins altered by the WD. Latent variable projections from PLS-DA on liver proteomics data (*Figure 4A*) showed that the WD group had a protein abundance profile significantly different from CN ($p<10^{-6}$, Hotelling's $T^2$). The effect of I3A was dose-dependent; whereas the WD-50 group was not significantly different from the WD group, the WD-100 group was significantly different from the WD group ($p<0.001$). The PLS-DA projections for WD-100 were closer to CN than WD, but still significantly different from CN. PCA of the proteomics data showed similar trends (*Figure 4—figure supplement 1*).

Clustering and silhouette analysis on significant (differentially abundant) proteins with variable importance in projection (VIP) scores >1.2 in the PLS-DA projections identified three clusters of proteins based on their abundance profiles. Proteins in cluster 1 had lower abundance in the WD group compared to CN. This trend was reversed in the WD-100 group (*Figure 4B*). Proteins in cluster 2 had lower abundance in the WD group compared to CN, but this trend was not reversed in the WD-100 group. Proteins in cluster 3 had higher abundance in the WD group compared to CN. These proteins had similar abundance in the WD-100 group and CN. The protein abundance profiles were similar between WD-50 and WD groups in all three clusters. These results suggest that the I3A treatment partially reversed the diet-induced alterations in the liver proteome in a dose-dependent manner, similar to the trend observed for the liver metabolome.

We performed an enrichment analysis to identify liver biochemical pathways significantly altered by diet and I3A treatment. In total, 454 quantified proteins were associated with 276 KEGG pathways. Of these, 22 proteins were differentially abundant in WD mice compared to CN. These proteins were associated with six significantly enriched pathways (*Figure 4C*). A comparison of the WD-100 and WD groups identified 52 differentially abundant proteins that were associated with five significantly

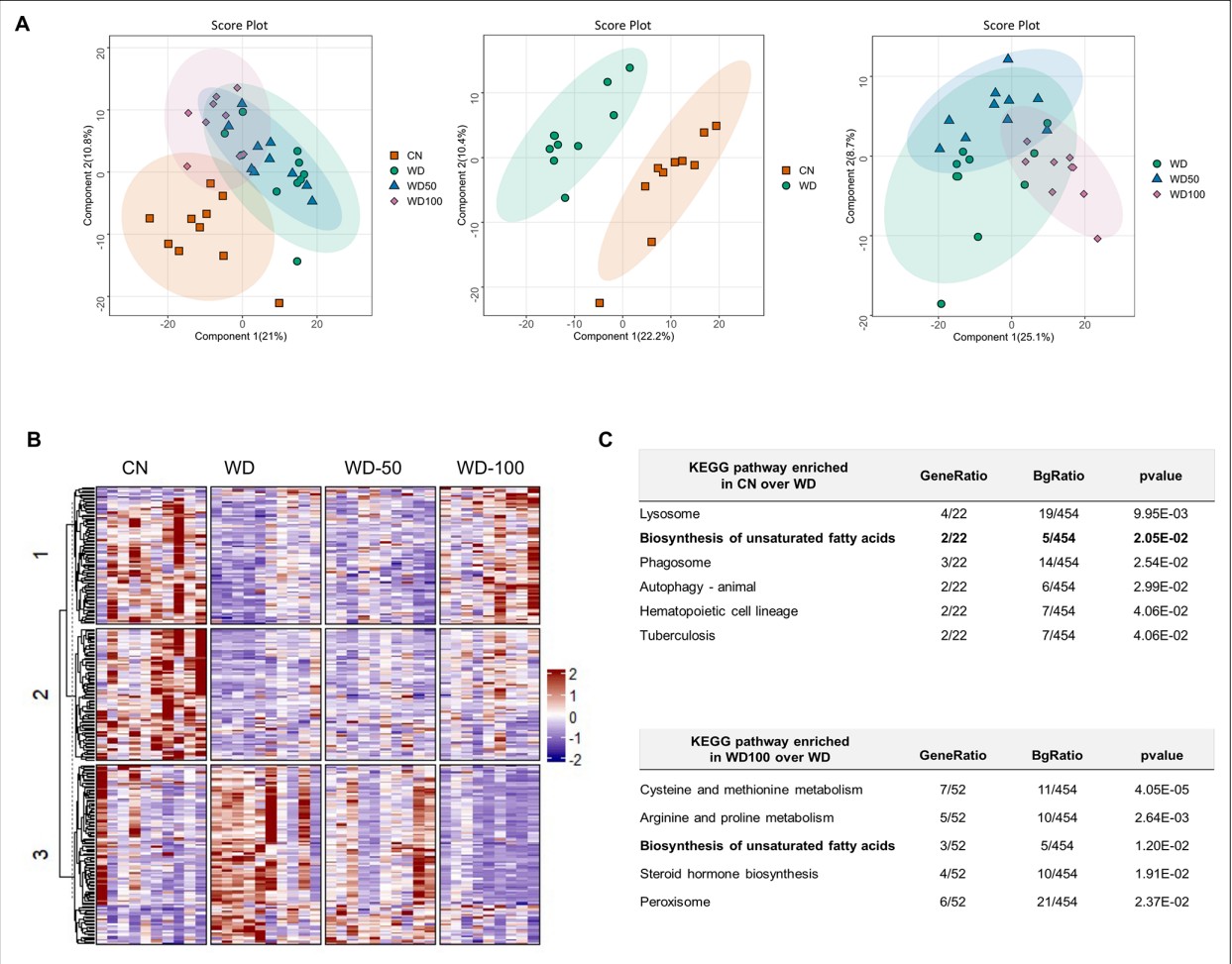

**Figure 4.** Indole-3-acetate (I3A) administration partially reverses diet-induced proteome alterations in the liver. (**A**) Scatter plots of latent variable projections from PLS-DA of confidently identified proteins. Comparison of all four experimental groups (left panel), control mice (CN) vs Western diet (WD) group (middle panel), and WD vs. WD-50 and WD-100 groups (right panel). (**B**) Heatmap of significant proteins having variable importance in projection score >1.2. The proteins were clustered using k-means. (**C**) Pathway enrichment analysis of significant proteins differentially abundant in CN vs. WD comparison (upper panel) and WD-100 vs. WD comparison (lower panel). GeneRatio divides the number of significantly altered proteins that are in the pathway by the total number of significantly altered proteins. BGRatio divides the number of proteins that are in the pathway by the number of all detected proteins. The p-value was calculated using Fisher's exact test.

The online version of this article includes the following figure supplement(s) for figure 4:

**Figure supplement 1.** Score plots of the first two principal components for all four experimental groups (left panel), control mice (CN) vs. Western diet (WD) group (middle panel), and WD vs. WD-50 and WD-100 groups (right panel).

enriched pathways. One metabolic pathway, biosynthesis of unsaturated fatty acids, was common to both sets of enriched pathways.

We used targeted proteomics to quantify fatty acid metabolism enzymes identified in the untargeted proteomics data (*Figure 5*). Platelet glycoprotein 4 (CD36), a major fatty acid uptake protein, was elevated in the WD group compared to CN. The abundance of CD36 was decreased, but not significantly, in the WD-100 group compared to WD (*Figure 5A*). The abundance of fatty acid synthase (Fasn) was not significantly different between the WD group and CN but was decreased in the WD-100 group compared to the WD group (*Figure 5A*). Acetyl-CoA carboxylase-2 (Acab), which regulates mitochondrial β-oxidation, was decreased in the WD group compared to CN, but was not significantly different between the WD-100 group compared to WD (*Figure 5B*). Enzymes catalyzing mitochondrial β-oxidation, including long-chain acyl-CoA dehydrogenase (Acadl), medium-chain acyl-CoA dehydrogenase (Acadm), short-chain acyl-CoA dehydrogenase (Acads), and hydroxyacyl-CoA dehydrogenase (Hadh) trended higher in the WD group compared to CN, although only Hadh showed a statistically

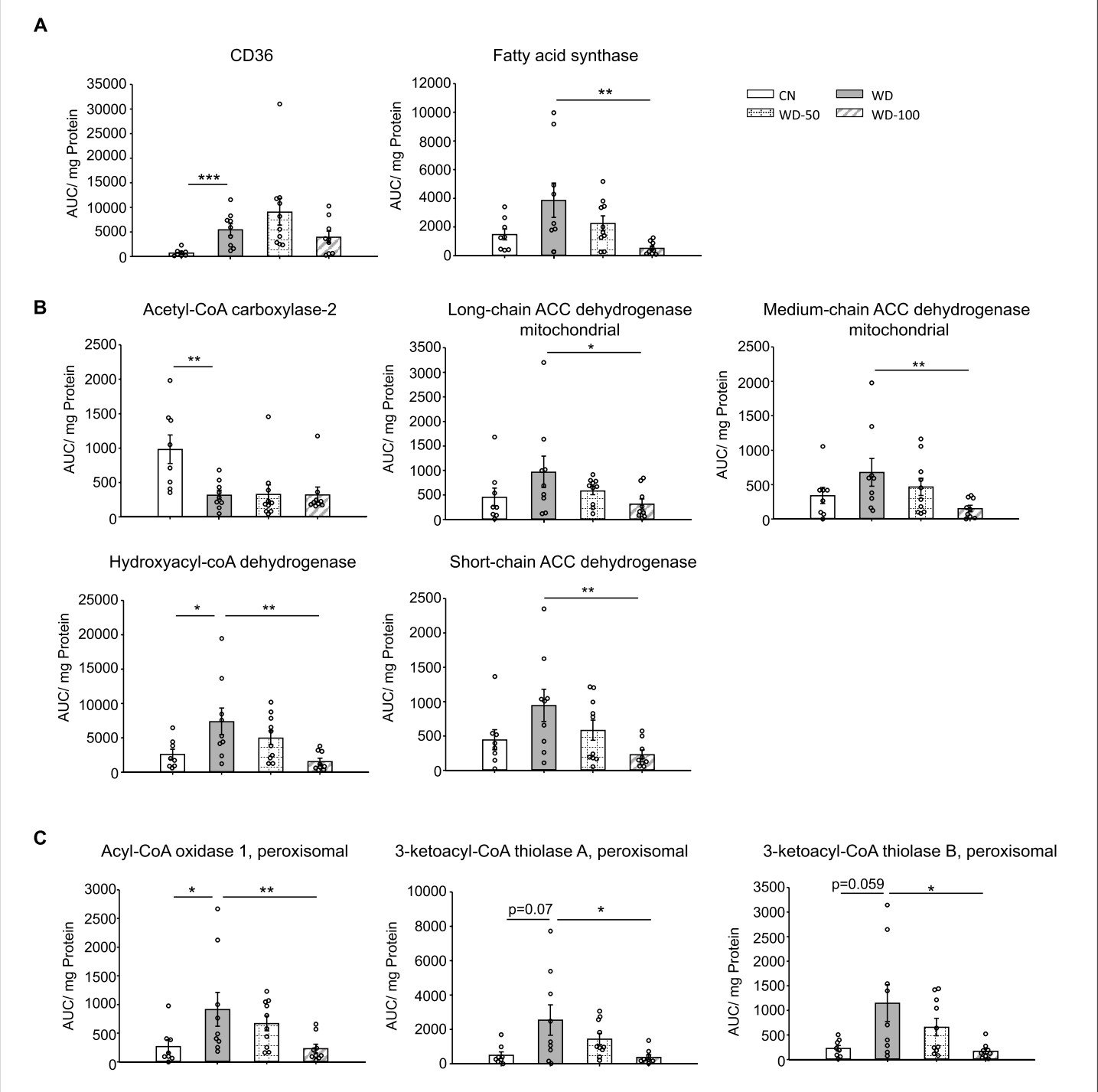

**Figure 5.** Indole-3-acetate (I3A) administration reduces the levels of enzymes in fatty acid transport, de novo lipogenesis, and β-oxidation. (**A**) Abundance of fatty acid translocase (CD36) and fatty acid synthase. (**B**) Mitochondrial and (**C**) peroxisomal fatty acid oxidation enzymes. Data shown are mean ± SEM. *: p<0.05, **: p<0.01, ***: p<0.001 using Wilcoxon rank sum test.

The online version of this article includes the following figure supplement(s) for figure 5:

**Figure supplement 1.** Targeted proteomic analysis of mouse liver tissue.

significant difference (*Figure 5B*). All four enzymes were significantly decreased in the WD-100 group compared to WD. A similar trend was observed for peroxisomal β-oxidation enzymes acyl-CoA oxidase 1 (Acox1) and peroxisomal 3-ketoacyl-CoA thiolases (Acaa1a and Acaa1b, *Figure 5C*). These results suggest that WD feeding led to increased cellular uptake, transport into mitochondria, and β-oxidation of fatty acids, while administration of I3A reduced fatty acid synthesis and β-oxidation. As β-oxidation generates reactive oxygen species (ROS), we also performed a targeted analysis of two antioxidant enzymes detected by the untargeted proteomics experiments, catalase (CatB) and gluta-thione peroxidase 1 (Gpx1), and administration of I3A reduced the levels of both enzymes compared to WD (*Figure 5—figure supplement 1A*). Previous in vitro studies in hepatocytes have shown that AhR activation upregulates hepcidin (Hamp) (*Hamano et al., 2018*; *Eleftheriadis et al., 2016*), a secreted liver protein that regulates iron absorption. Mice fed the WD showed significantly reduced Hamp abundance in the liver, which was dose-dependently increased by I3A treatment (*Figure 5—figure supplement 1A*). Another iron carrier protein, serotransferrin, showed an opposite trend; its abun-dance significantly increased in the WD group compared to CN, and dose-dependently decreased by I3A treatment (*Figure 5—figure supplement 1B*).

## I3A suppresses macrophage inflammation in an AMPK- but not AhR-dependent manner

Since our previous study found that the metabolic effects of I3A in hepatocytes depend on the AhR, we tested if this was also the case in macrophages. The expression of AhR in RAW 264.7 macrophages is very low compared to murine AML12 hepatocytes (*Figure 6—figure supplement 1A*). Treatment with AhR inhibitor CH223191 did not alter I3A's effect on reducing palmitate and LPS-induced *Tnfa* and *Il-β* expression in RAW 264.7 macrophages (*Figure 6—figure supplement 1B, C*), indicating that the anti-inflammatory effect of I3A in macrophages likely does not require AhR activity.

Given that AMPK is a well-known signaling mediator in lipid metabolism and inflammation (*Muse et al., 2004*; *Sag et al., 2008*; *Day et al., 2017*), and has been shown to play a role in the progression of NAFLD (*Smith et al., 2016*; *Herzig and Shaw, 2018*), we investigated if the levels of AMPK and p-AMPK were altered in the different treatment groups. Both p-AMPK and AMPK were significantly downregulated in livers of the WD group compared to CN (*Figure 6A*). Administration of I3A reversed this effect in a dose-dependent manner and increased both AMPK and p-AMPK (*Figure 6B*).

We tested the role of AMPK in RAW 264.7 macrophages in vitro. Palmitate and LPS treatment significantly reduced p-AMPK levels by 50% compared to vehicle control but did not affect total AMPK (*Figure 6C and D*). Treatment with I3A upregulated p-AMPK to baseline levels without altering total AMPK (*Figure 6C and D*). Palmitate and LPS also led to an increase in *Tnfa* and *Il-1β* expression (*Figure 6E*), while activation of AMPK with 5-aminoimidazole-4-carboxamide ribonucleotide (AICAR) attenuated the palmitate and LPS-stimulated increase in *Tnfa* and *Il-1β* expression by 30% and 80%, respectively (*Figure 6E*). These results suggest that upregulation of p-AMPK modulates the expres-sion of *Tnfa* and *Il-1β* in RAW 264.7 macrophages.

We next investigated if the anti-inflammatory effect of I3A in RAW macrophages that we previ-ously reported (*Krishnan et al., 2018*) depends on AMPK activation. We used siRNA to knock down prkaa1, the gene encoding AMPKα1, the main form of AMPKα in murine macrophages (*Yang et al., 2010*), and measured its effect on pro-inflammatory cytokine expression. The knockdown efficiency was ~50% for mRNA (*Figure 6—figure supplement 2A*), ~40–50% for AMPK protein, and ~50–60% for p-AMPK (*Figure 6—figure supplement 2B, C*), and the knockdown was stable for up to 96 hr at both mRNA and protein levels. In cells transfected with the non-targeted control siRNA (*Figure 6F*, left panel), palmitate and LPS increased *Tnfa* and *Il1b* expression by ~10- and 250-fold, respectively (*Figure 6F*), and I3A significantly downregulated this by ~30% and 50% for *Tnfa* and *Il1b*, respectively, relative to the DMF control. In contrast, I3A did not significantly modulate *Tnfa* expression in prkaa1 siRNA-transfected cells stimulated with palmitate and LPS, and only reduced *Il1b* expression by ~20% (*Figure 6F*) relative to the DMF control. These results suggest that I3A's anti-inflammatory effect in RAW 264.7 macrophages depends on AMPK activation.

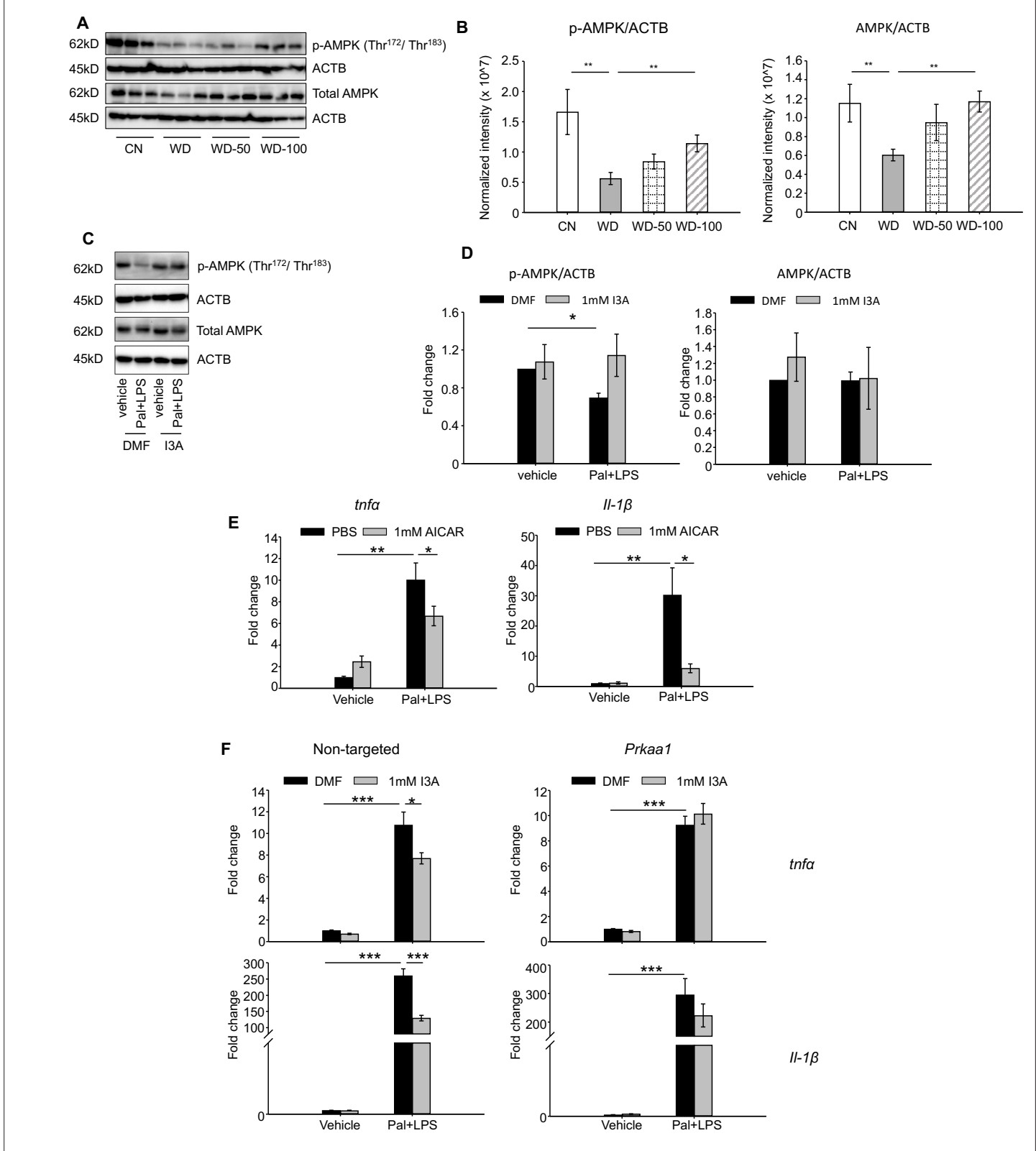

**Figure 6.** Indole-3-acetate (I3A) modulates AMP-activated protein kinase (AMPK) phosphorylation and suppresses RAW264.7 macrophage cell inflammation in an AMPK-dependent manner. (**A, B**) I3A administration reverses Western diet (WD)-induced reduction in liver p-AMPK and AMPK. (**A**) Levels of p-AMPK and AMPK in liver tissue at week 16 as determined by western blot analysis. (**B**) Ratios of p-AMPK (left panel) and AMPK (right panel) to β-actin. The ratios were determined based on the p-AMPK and AMPK band intensities quantified using Image Lab (Bio-Rad) and normalized to the

*Figure 6 continued on next page*

*Figure 6 continued*

loading control (β-actin). Data shown are mean ± SEM. **: p<0.01 using Wilcoxon rank sum test. (**C**) Expression levels of p-AMPK and total AMPK in RAW 264.7 macrophages pre-treated with either I3A or vehicle (DMF) control followed by stimulation with palmitate and LPS, determined by western blot analysis. (**D**) Fold-changes in p-AMPK and total AMPK. Fold-changes were calculated relative to the DMF and no palmitate and LPS stimulation condition. The band intensities were quantified and normalized to loading control (β-actin) by using Image Lab (Bio-Rad). (**E**) Expression levels of *Tnfa* and *Il1b* in RAW 264.7 cells treated with p-AMPK activator 5-aminoimidazole-4-carboxamide ribonucleotide (AICAR), followed by stimulation with palmitate and LPS. (**F**) Expression levels of *Tnfa* (top row) and *Il1b* (bottom row) in RAW 264.7 cells transduced with non-targeted control siRNA (left panels) or *Prkaa1* siRNA (middle panels), pre-treated with I3A, and then stimulated with palmitate and LPS. Data shown are mean ± SEM from three independent cultures with three biological replicates. *: p<0.05, **: p<0.01, ***: p<0.001 using Student's t-test.

The online version of this article includes the following source data and figure supplement(s) for figure 6:

**Source data 1.** Western blot analysis of p-AMPK and AMPK levels in mice liver tissues used in *Figure 6A*.

**Source data 2.** Western blot analysis of p-AMPK and AMPK levels in RAW264.7 cells used in *Figure 6C*.

**Figure supplement 1.** Anti-inflammatory effects of I3A on RAW264.7 cells is independent of AhR.

**Figure supplement 1—source data 1.** Western blot analysis of AhR level in AML12 and RAW264.7 cells used in *Figure 6—figure supplement 1A*.

**Figure supplement 2.** Knockdown of AMP-activated protein kinase (AMPK) by siRNA transfection.

**Figure supplement 2—source data 1.** Western blot analysis of p-AMPK and AMPK levels in RAW264.7 cells treated with siRNAs used in *Figure 6—figure supplement 2B*.

**Figure supplement 3.** When mice are fed with a Western diet (WD) (top panel), triglycerides (TG) and free fatty acids (FFAs) accumulate in the liver due to increased uptake of fatty acids.

## Discussion

Based on our previous finding that I3A modulates inflammatory cytokine expression in macrophages and fatty acid metabolism in hepatocytes (*Krishnan et al., 2018*), we hypothesized that I3A could mitigate liver steatosis and other NAFLD features in vivo. Using a mouse model of WD-induced NAFLD (*Asgharpour et al., 2016*), we show that oral administration of I3A protects against liver injury in a dose-dependent manner (*Figure 1D*), attenuates liver triglyceride (TG) accumulation (*Figure 1E*), and reduces steatosis, hepatocyte ballooning, and lobular inflammation (*Figure 1F and G*), suggesting that administration of I3A altered both metabolic and inflammation pathways in the liver. As expected, WD induced significant additional weight gain. Treatment with I3A had negligible impact on the weight gain and did not reduce food intake.

Production of pro-inflammatory cytokines such as *Tnfa*, *Il1b*, *Ccl2*, and *Il6* by resident macrophages (Kupffer cells) signals recruitment and infiltration of monocytes into the liver. These cytokines also promote the differentiation of monocytes into pro-inflammatory macrophages, which in turn exacerbates dysregulation of hepatic lipid metabolism (*Seki and Schwabe, 2015*; *Braunersreuther et al., 2012*). Consistent with the lobular inflammation score, there was a significant increase in the expression of inflammatory cytokines in livers of the WD group (*Figure 2A*). Remarkably, treatment of WD-fed mice with I3A dose-dependently reduced the expression of every cytokine in the 13-member panel, including IL-10. The downregulation of IL-10 in I3A-treated mice could be a result of reduced hepatic inflammation (i.e. feedback regulation of IL-10 production). Whether the decrease in inflammatory cytokine production is due to I3A's anti-inflammatory effects in liver immune cells (*Krishnan et al., 2018*) or due to modulation of lipid metabolism in hepatocytes warrants further investigation.

Studies have shown that the alteration of bile acid metabolism could be a biomarker for NAFLD (*De Juan and Segura, 2021*). Mice in the WD group had a lower total concentration of liver bile acids and higher proportion of primary bile acids compared to CN (*Figure 2B*). The latter trend is consistent with previous studies comparing bile acid profiles between NASH patients and healthy controls (*Puri et al., 2018*). Compared to our previous in vitro results, the effect of I3A treatment on liver bile acids was modest, possibly due to regulatory mechanisms present in vivo. Moreover, CDCA is metabolized into muricholic acids in mice which are absent in humans and have different affinities for bile acid receptors compared to CDCA and lithocholic acid, a CDCA-derived secondary bile acid. Knockout mouse models that lack the bile acid metabolizing enzymes Cyp2c70/Cyp2a12 have bile acid pools closer to humans (*de Boer et al., 2020*; *Ueda et al., 2022*). Future studies utilizing these models could help elucidate the physiological impact of I3A on liver bile acids and their role in fatty liver disease.

The observation that the fecal microbial community composition was not significantly altered by I3A administration suggests that the hepatoprotective effects of I3A are likely due to the metabolite's direct action on the host, rather than the gut microbiota composition. One caveat to this interpretation is the shallow resolution of 16S rRNA sequencing. It is possible that I3A administration altered the microbiota composition at the species level. It is also possible that I3A could affect the metabolism of the microbiota without altering its composition. A recent study (*Lobel et al., 2020*) showed that a diet containing high levels of methionine and cysteine protected mice against chronic kidney disease by reducing uremic toxins through post-translational modification of gut bacterial enzymes. To test the possibility that I3A administration altered the intestinal metabolome, we performed untargeted metabolomics experiments on fecal samples. The results showed no significant impact of I3A on the fecal metabolomes of WD mice. While further studies are warranted to fully elucidate the impact of I3A administration on the microbiome and microbial gene expression in the intestine, these results suggest that the partial reversal of WD-induced changes in liver metabolome and proteome by I3A treatment is due to the metabolite's direct action on the liver.

Analysis of enzymes in lipid transport, synthesis, and mobilization showed that WD and I3A impacted different facets of lipid metabolism. Targeted proteomics detected a sixfold increase in fatty acid translocase CD36 in the WD group compared to CN, suggesting that the elevation of liver FFAs in the WD group (*Figure 2A*) could be due to increased uptake. I3A did not affect CD36 abundance, but significantly decreased Fasn by 86%, the rate limiting enzyme in de novo lipogenesis. This suggests that the significant decrease in liver TG in the WD-100 group could reflect a downregulation of de novo lipid synthesis in the liver.

Liver fatty acid oxidation (FAO) enzymes also showed different responses to WD and I3A. Acab is a mitochondrial regulatory enzyme that produces malonyl-CoA, which inhibits carnitine palmitoyl-transferase 1 (CPT1) (*Abu-Elheiga et al., 2000*). Downregulation of Acab in the WD group compared to CN suggests increased CPT1 activity and hence elevated long-chain FA (LCFA) transport into the mitochondria (*Reddy and Hashimoto, 2001*). This interpretation is also supported by the observation that long-chain 3-hydoxy-acyl carnitines were elevated in the WD group compared to CN, which is consistent with a recent study reporting elevated serum C14 hydroxy-acylcarnitine in NAFLD patients (*Enooku et al., 2019*). Other mitochondrial FAO enzymes also showed a trend toward upregulation in the WD group compared to CN. Upregulation of FAO by WD was significant in the peroxisomes, the initial site of very LCFA oxidation. The literature is conflicted on FAO in human subjects with NAFLD or NASH. Depending on the study, enhanced (*Sunny et al., 2011*; *Dasarathy et al., 2011*; *Miele et al., 2003*), unchanged (*Petersen et al., 2016*; *Kotronen et al., 2009*), or decreased (*Croci et al., 2013*) FAO has been reported. The different results may reflect varying severity of the disease (degree of steatosis or steatosis vs. NASH) and variations in FAO capacity across individual subjects. One study (*von Loeffelholz et al., 2017*) showed that the expression of genes in peroxisomal and mitochondria β-oxidation was higher in patients with more severe steatosis compared to patients with less sever steatosis or healthy control. Administration of I3A significantly reduced the abundance of both mitochondrial and peroxisomal FAO enzymes. Increased FAO and concomitant ROS generation can overwhelm cellular antioxidant defenses, inducing oxidative stress. Although there is a lack of consensus regarding FAO, studies consistently report elevated markers of oxidative stress in human subjects with steatosis (*Palmieri et al., 2006*; *Del Ben et al., 2014a*; *Del Ben et al., 2014b*). In this regard, I3A may protect from oxidative stress by reducing FAO. This is consistent with our observations that I3A administration reversed the WD-induced accumulation of long-chain 3-hydroxy-acylcarnitines and that the changes in abundance of FAO enzymes were upstream of 3-hydroxy-acyl-CoAs. Oxidative stress-induced enzymes CAT and GPx were upregulated in the WD group compared to CN and downregulated in the WD-100 group compared to the WD group. It should be noted that all the biochemical parameters related to lipid metabolism were assessed at termination of the mouse experiments. It is possible that the downregulation of FAO enzymes in I3A-treated mice reflects a response to reduced lipid accumulation. Further longitudinal studies are warranted to determine the (likely) dynamic effects of I3A on the liver lipid metabolism.

Recent studies have demonstrated that tryptophan-derived gut bacterial metabolites, including I3A, are AhR agonists in intestinal epithelial cells (*Jin et al., 2014*), hepatocytes (*Krishnan et al., 2018*), and immune cells (*De Juan and Segura, 2021*). In our previous study (*Krishnan et al., 2018*), we observed that I3A decreased the expression of Fasn and its transcriptional regulator SREBP-1c in

hepatocytes in an AhR-dependent manner. In the current study, we again found that Fasn was down-regulated by I3A treatment. Additionally, we observed a dose-dependent upregulation of hepcidin in mice treated with I3A. Previously, tryptophan-derived metabolites indoxyl sulfate and kynurenine have been shown to induce hepcidin expression in HepG2 cells in an AhR-dependent manner (*Hamano et al., 2018*; *Eleftheriadis et al., 2016*). Hamp is a master regulator of systemic iron homeostasis that binds to ferroportin and inhibits the absorption of dietary iron and efflux of iron from liver resident macrophages. Iron overload is common among NAFLD patients (*Datz et al., 2017*), and the excess iron along with inflammation can induce hepcidin in these patients (*Vela, 2018*; *Zhou and Qiu, 2022*; *Senates et al., 2011*; *Mitsuyoshi et al., 2009*). However, hepcidin gene expression decreases when NAFLD progresses to NASH (*Mitsuyoshi et al., 2009*), suggesting that disease progression is linked to impaired regulation of iron metabolism. Hepcidin knockout in a mouse model of NAFLD ameliorated steatosis while exacerbating fibrosis (*Lu et al., 2016*). In contrast, hepcidin overexpression decreased choline-deficient diet-induced steatosis, inflammation, and fibrosis in a mouse model of NASH (*Chen et al., 2022*). Whether I3A induction of hepcidin ameliorates iron overload and how this could contribute to I3A's protective role needs to be investigated.

Another set of WD-induced metabolic alterations that were reversed by I3A treatment and likely mediated through the AhR are restorations of tryptophan metabolite levels. Studies in hepatocytes have shown that AhR knockdown downregulates tryptophan 2,3-dioxygenase, kynurenine 3-monooxygenase, and kynureninase, enzymes that direct flux into the kynurenine branch of tryptophan metabolism. Another study reported that kynurenine activation of the AhR upregulated expression of indoleamine 2,3-dioxygenase 2 in dendritic cells (*Li et al., 2016*). In the present study, I3A treatment of WD-fed mice directed flux of tryptophan into kynurenic, anthranilic, and xanthurenic acids, reducing flux toward niacin and nicotinamide. Given that kynurenine, kynurenic acid, anthranilic acid, and xanthurenic acid are all AhR agonists, this could have further enhanced AhR activation by I3A treatment.

In macrophages, however, I3A's effects were independent of AhR activity, as addition of the AhR inhibitor did not alter the effects (*Figure 6—figure supplement 1*). Studies in HFD-fed mice showed dysregulation of liver AMPK activity and its association with increased lipid accumulation (*Muse et al., 2004*; *Sag et al., 2008*; *Wu et al., 2007*). In vitro, LPS-activated macrophage showed upregulation of heme oxygenase-1, an AMPK-regulated protein, upon I3A exposure (*Ji et al., 2020*). Based on these reports, we investigated if AMPK was involved in mediating the response to I3A. We show that both liver AMPK expression and phosphorylation were reduced in WD-fed mice relative to CN, and that administration of I3A rescued both AMPK expression and phosphorylation in a dose-dependent manner (*Figure 6*). Using an in vitro model, we demonstrate a similar AMPK dependence of I3A's anti-inflammatory effect in murine macrophages (*Figure 6*). Whether AMPK signaling plays a role in mediating I3A's effects in hepatocytes in vivo and whether this is important relative to I3A's activation of the AhR warrants further investigation.

In summary, we have shown that oral administration of a microbiota-derived tryptophan metabolite, I3A, in WD-fed mice alleviates diet-induced liver steatosis and inflammation, even when the mice were continued on WD. These hepatoprotective effects occurred without significant alterations in the gut microbiome composition and metabolome profiles, suggesting that I3A acted directly on host cells. Untargeted LC-MS experiments showed activation of several AhR-regulated pathways, suggesting that I3A's effects are partially mediated through this nuclear receptor. Additionally, in vivo results show a correlation between AMPK phosphorylation and the efficacy of I3A, while our in vitro studies with RAW macrophages show that AMPK mediates I3A's attenuation of pro-inflammatory cytokine expression. While our results do not rule out the involvement of additional signaling pathways in both hepatocytes and macrophages, or interactions between the liver and other tissues (e.g. adipose tissue), the data nevertheless strongly point to I3A directly modulating lipid metabolism and inflammatory cytokine production in the liver.

Recently, several other studies reported on the protective effects of I3A in various mouse models (*Zhang et al., 2022*; *Ji et al., 2019*). However, there are significant differences between the other studies and our work. The prior studies modeled steatosis using short-term (<4 weeks) HFD exposure, whereas we modeled the later stages of NAFLD (i.e. with steatosis and inflammation) utilizing long-term (16 weeks) exposure to WD and SW. Moreover, in contrast to the other studies, I3A was administered concurrently with WD and SW for the last 8 weeks of the study. Despite the differences in the

study design, the common conclusion from the above studies and our work is that I3A administration decreases hepatic TG. In addition, our study shows that administration of I3A significantly decreases hepatic inflammation and lipid metabolism with minimal modulation of the gut microbiome composition and metabolic function. Mechanistically, we also show that I3A's anti-inflammatory effects in RAW 264.7 macrophages are mediated through activation of AMPK, whereas its effects in hepatocytes are mediated through the AhR (*Krishnan et al., 2018*). Thus, demonstrating that I3A elicits protective effects in NAFLD through two signaling pathways in different cell types is a novel and significant finding from our study.

Based on our data, we propose a model describing I3A's effects in the liver (*Figure 6—figure supplement 3*). While additional studies are warranted to investigate the above-mentioned possibilities, the remarkable potency of I3A in alleviating steatosis and inflammation in a therapeutic model clearly demonstrates the potential for developing I3A as an inherently safe treatment option for NAFLD. To this end, preclinical studies should be conducted to evaluate delivery options such as botanical extracts and probiotics, while human subject studies should be performed to determine safety, tolerability, and side effects of I3A. Prospectively, other tryptophan-derived microbiota metabolites as well as diets that result in enhanced production of these metabolites in the GI tract could also be investigated for protection against WD-induced hepatic steatosis and inflammation.

## Materials and methods
### Study design
The overall objective of the study was to investigate if I3A can alleviate diet-induced liver steatosis and inflammation in vivo. We induced these NAFLD features in mice by feeding the animals a WD and SW for 8 weeks. The drinking water for these mice was then supplemented with low and high doses of I3A, while continuing the WD for another 8 weeks. The phenotypic changes were assessed by measuring body weight gain, serum ALT, hepatic inflammatory cytokines, and histopathological examination of liver tissue. To investigate potential mechanisms of action, we performed a series of omics analyses on fecal material (metagenomics and metabolomics) and liver tissue (metabolomics and proteomics). To determine cell-type-specific signaling mediating I3A effects in macrophages, we used an in vitro cell culture model. The histopathological scoring of liver sections was blinded; all other analyses were not blinded. Sample processing and statistical analysis were performed concurrently on treatment and control groups using identical methods. Numbers of replicates and outcomes of statistical tests are indicated in the figure legends.

### Materials
RAW 264.7 cells were purchased from ATCC (Manassas, MA, USA). Cells were authenticated and tested for mycoplasma contamination using the CellCheck 19 Plus test (ATCC). Dulbecco's Modified Eagle Medium (DMEM), penicillin/streptomycin, and LPS (from *Salmonella minnesota*) were purchased from Invitrogen (Carlsbad, CA, USA). Fetal bovine serum (FBS) was purchased from Atlanta Biologicals (Flowery Branch, GA, USA). All FFA chemicals, AICAR, and the AhR inhibitor CH-223191 were purchased from Millipore Sigma (St. Louis, MO, USA). I3A sodium salt was purchased from Cayman chemicals (Ann Arbor, MI, USA). All other chemicals were purchased from VWR (Radnor, PA, USA) or Millipore Sigma unless otherwise specified.

### Animal experiments
Male B6 129SF1/J mice at 6 weeks of age were obtained from Jackson Laboratories (Bar Harbor, ME, USA). Mice were acclimatized to the animal facility for 1 week. At the start of the experiment, mice were randomly divided into four groups (n=10 for each group). Three of the four groups were fed ad libitum a WD with 40% kcal from fat and containing 0.15% cholesterol (D12079B, Research Diets) and an SW solution (23.1 g/l d-fructose+18.9 g/l d-glucose) as previously described (DIAMOND model; *Asgharpour et al., 2016*). After 8 weeks, the three groups of WD-fed mice were randomly selected for treatment with vehicle (WD group) or low (WD-50 group) or high dose (WD-100 group) of I3A. The WD group was fed WD and drank SW. The WD-50 and WD-100 groups were fed WD and drank SW containing, respectively, 50 or 100 mg I3A per kg body weight (corresponding to 0.5 or 1 mg/ml, respectively). The treatments were continued for another 8 weeks. Water bottles containing SW and

I3A or only SW were changed every other day. A fourth group of mice was fed a low-fat control diet (D12450B, Research Diets) and normal water for 16 weeks (CN group) (*Figure 1—figure supplement 1A*). Mice belonging to the same treatment group were housed together (5 mice/cage). Mice were maintained on 12:12 hr light-dark cycles. All procedures were performed in accordance with Texas A&M University Health Sciences Center Institutional Animal Care and Use Committee guidelines under an approved animal use protocol (AUP #2017-0145).

## Sample collection

Fecal pellets were collected every other week prior to I3A treatment, right before I3A treatment, every week during I3A treatment, and on the last day of the experiments (*Figure 1—figure supplement 1B*). All fecal pellets were flash-frozen in liquid nitrogen after collection. Blood samples from the submandibular vein were taken right prior to I3A treatment, every other week during I3A treatment and on the last day of the experiments. The blood samples were centrifuged at 4000 × *g* for 15 min at 4°C to obtain serum. At the end of the experiment, the mice were sacrificed by euthanasia. The liver was quickly excised and rinsed with 10× volume of ice-cold PBS. The right medium lobe of the liver was fixed in 10% neutral formalin for histological analysis. A small piece from right lateral lobe was stored in RNAlater (Sigma-Aldrich, St. Louis, MO, USA) using the manufacturer's protocol. The remaining liver tissue samples were flash-frozen in liquid nitrogen, homogenized in HPLC-grade water, lyophilized to a dry powder, and stored at –80°C until further processing.

## Histological analysis

Formalin-fixed liver were embedded in paraffin, sectioned (5 μm), and stained with H&E through VWR histological services (Radnor, PA, USA). Liver histology sections were evaluated by an expert pathologist at Texas A&M University who was blinded to the treatment conditions. Histology was assessed using the NASH CRN (*Kleiner et al., 2005*) and fatty liver inhibition of progression (FLIP) consortia criteria (*Bedossa and FLIP Pathology Consortium, 2014*).

## Serum ALT and hepatic TG measurement

ALT was measured in serum samples using a commercial ELISA kit (Cayman Chemical Company). Liver TG were quantified using a commercial Colorimetric Assay Kit (Cayman Chemical Company). Briefly, lyophilized liver samples were weighted and lysed in diluted NP-40 buffer. After centrifugation at 10,000 × *g* and 4°C for 15 min, the supernatant was stored on ice for quantification of TG. A small amount of lyophilized liver was used for DNA isolation using a DNA Miniprep Kit (Zymo Research, Irvine, CA, USA). The TG concentrations (mg/dl) were normalized to tissue DNA contents (μg).

## Fecal microbiome analysis

Microbial DNA was extracted from homogenized fecal material using the PowerSoil DNA Extraction Kit (QIAGEN, Carlsbad, CA, USA). The V4 region of 16S rRNA was sequenced on a MiSeq Illumina platform (*Kozich et al., 2013*) at the Microbial Analysis, Resources, and Services (MARS) core facility (University of Connecticut). Illumina sequence reads were quality filtered, denoised, joined, chimera filtered, aligned, and classified using mothur (v1.40.4) following the MiSeq SOP pipeline. The SILVA database (*Quast et al., 2013*) was used for alignment and classification of the operational taxonomic units (OTUs) at 97% similarity. The taxonomic dissimilarities between different treatment groups were calculated using the Bray-Curtis dissimilarity metric and visualized on non-metric multidimensional scaling plots. Analysis of similarities was used for statistical comparison of microbiome compositions between treatment groups. The differential abundances of OTUs between two different groups were determined using LEfSe (*Segata et al., 2011*). Fisher's index was calculated to estimate the alpha diversity. Sequencing data has been deposited in the Sequence Read Archive (SRA) database (PRJNA1029996).

## Fecal and serum metabolite extraction

Metabolites were extracted from fecal samples as described previously (*Yang et al., 2020*). Briefly, fecal material was weighed, homogenized, and extracted twice with chloroform/methanol/water. The aqueous phase from the two extractions were combined, lyophilized, and stored at –80°C. Samples were reconstituted in 100 μl methanol/water (1:1, vol/vol) prior to LC-MS analysis. Serum metabolites

were extracted with ice-cold methanol (1:4 serum:methanol). Samples were centrifuged twice at 15,000 × g for 5 min at 4°C. The supernatant was stored at –80°C until LC-MS analysis.

## Liver metabolite and protein extraction

Lyophilized liver samples were weighted and homogenized using soft tissue homogenization beads (Omni International) on a bead beater (VWR) with 1 ml ice-cold methanol/water (91:9, vol/vol). The samples were homogenized for 1 min, incubated on ice for 5 min, and centrifuged at 12,000 × g for 10 min at 4°C. The supernatant was transferred into a new sample tube through a 70 µm cell strainer. Ice-cold chloroform was added into the tube to obtain a final solvent ratio of 47.6/47.6/4.8% methanol/chloroform/water (*Dettmer et al., 2011*). After vigorous mixing, the samples were frozen in liquid nitrogen and thawed at room temperature. The freeze-thaw cycle was carried out three times. Samples were centrifuged at 15,000 × g for 5 min at 4°C. The supernatant and pellet were each transferred into a new sample tube for metabolite and protein analysis, respectively.

For metabolite analysis, 1 ml of HPLC water was added to the supernatant and centrifuged at 10,000 × g for 5 min at 4°C to obtain phase separation. The upper and lower phases were collected separately and filtered through 0.2 µm filters into new sample tubes. The filtered samples were dried to pellets with a lyophilizer and stored at –80°C until further analysis. The upper and lower phases were reconstituted in 100 µl methanol/water (1:1, vol/vol) and 200 µl methanol, respectively, prior to LC-MS analysis. For protein analysis, the pellet was solubilized in 650 µl extraction buffer (0.5% SDS, 1% vol/vol β-mercaptoethanol, and 1 M Tris-HCl, pH = 7.6) and 650 µl TRIzol reagent (Thermo Fisher Scientific, Waltham, MA, USA) and incubated at 37°C for 1 hr. The sample was centrifuged at 14,000 × g for 15 min at 4°C to obtain phase separation. One ml of ice-cold acetone was mixed into the bottom protein layer. Following an overnight incubation at –20°C, the samples were centrifuged at 14,000 × g for 15 min at 4°C. The supernatant was discarded, and the pellet was washed 3× with 1 ml ethanol. The protein pellet was then lyophilized and stored at –80°C until further analysis.

## Untargeted metabolomics

Untargeted LC-MS experiments were performed on a Q Exactive Plus orbitrap mass spectrometer (Thermo Fisher Scientific, Waltham, MA, USA) coupled to a binary pump HPLC system (UltiMate 3000, Thermo Fisher Scientific, Waltham, MA, USA) at the Integrated Metabolomics Analysis Core (IMAC) facility of Texas A&M University as previously described (*Yang et al., 2020*). Chromatographic separation was achieved on a reverse phase column (Synergi Fusion 4 µm, 150 mm × 2 mm, Phenomenex, Torrance, CA, USA) using a gradient method (*Supplementary file 1*). Sample acquisition was performed Xcalibur (Thermo Fisher Scientific, Waltham, MA, USA). Raw data files were processed in Compound Discoverer (v3.0, Thermo Fisher Scientific, Waltham, MA, USA). Metabolite identification was performed by searching the features against mzCloud and ChemSpider. For comparing the levels of metabolites between samples, the area under the curve (AUC) determined in Compound Discoverer for each feature was normalized to the sum of the AUCs for all features. If there are multiple features annotated as the same metabolite, the feature with highest confidence score and highest AUC was selected to represent the metabolite. Proteomics data have been deposited to the ProteomeXchange Consortium via the PRIDE partner repository with the dataset identifier PXD046255.

## Liver bile acid analysis

Lyophilized liver samples were homogenized in 10 mM phosphate buffer at pH 6 (28 mg tissue per ml buffer) on a bead beater (VWR, Radnor, PA, USA) for 1 min. Samples were then centrifuged at 15,000 × g for 5 min at 4°C. The supernatant (200 µl) was mixed vigorously with 100 ng of d4-glycohenode-oxycholic acid internal standard, 20 µl of saturated ammonium sulfate, and 800 µl of acetonitrile, and then centrifuged at 15,000 × g for 5 min at 4°C. The supernatant was transferred into a new sample tube and dried with a vacufuge (Eppendorf, Hauppauge, NY, USA). Pellets were resuspended in 100 µl methanol/water (1:1, vol/vol), vortexed for 30 s, sonicated for 1 min, and then centrifuged at 15,000 × g and 4°C for 1 min. The supernatant was transferred into a new sample tube and analyzed by LC-MS.

Targeted LC-MS experiments were performed on a TSQ Altis triple quadrupole mass spectrometer (Thermo Fisher Scientific, Waltham, MA, USA) coupled to a binary pump UHPLC (Vanquish, Thermo Scientific) at the TAMU IMAC core facility. Chromatographic separation was achieved on a Kinetex 2.6 µm, 100 mm × 2.1 mm polar C18 column (Phenomenex, Torrance, CA, USA) using a gradient

method (*Supplementary file 2*). Scan parameters for target ions are listed in *Supplementary file 2*. Sample acquisition and data analysis were performed with Trace Finder 4.1 (Thermo Fisher Scientific, Waltham, MA, USA). The calculated bile acid concentrations were normalized to the level of spiked internal standard for each sample.

## Quantification of FFAs

Serum and liver FFAs were analyzed using targeted LC-MS experiments performed on a quadrupole-time of flight instrument (TripleTOF 5600+, AB Sciex, Framingham, MA, USA). Chromatographic separation was performed on a C18 column (Gemini 5 µm C18 110 Å Column, 250 mm×2 mm, 5 µm, Phenomenex) using the solvent gradient described in *Supplementary file 3*. The MS experimental parameters are listed in *Supplementary file 3*.

## Untargeted proteomics

Proteins were reduced, alkylated, and digested into peptides using information-dependent acquisition (IDA) on a quadrupole-time of flight instrument (TripleTOF 5600+) as previously described (*Manteiga and Lee, 2017*) with minor modifications (*Supplementary file 4*). Peptide ions detected in the IDA experiment were annotated using ProteinPilot (v5.0, AB Sciex) and further processed using an inhouse script written in MATLAB to associate each protein identified in the sample with a unique high-quality peptide (*Supplementary file 5*). The relative abundance of a protein was determined by quantifying the corresponding peptide peak's AUC in Multi-Quant (v2.0, AB) and normalizing the value by the sample's sum of all peptide AUCs.

## Targeted proteomics

A panel of 15 proteins having differential abundance in the untargeted proteomics data were selected for targeted measurements using product ion scans for representative peptides (*Supplementary file 6*). The relative abundance of a target protein was determined by quantifying the corresponding peptide peak's AUCs in MultiQuant and normalizing the value by the sample protein weight.

## Liver cytokine measurements

Lyophilized liver samples were weighted and homogenized with 500 µl lysis buffer (50 mM Tris, 150 mM NaCl, 1% Triton X-100, 1 mM EDTA, and 1% protease inhibitor cocktail, pH = 7.4) for 1 min on a bead beater (VWR, Radnor, PA, USA). Samples were centrifuged at 20,000 × $g$ for 15 min at 4°C. The lipid layer was removed by pipetting and the supernatant was transferred into a new tube. The centrifugation and lipid removal steps were repeated three times. The supernatant was transferred into a new tube and the total protein concentration was measured with the BCA protein assay (Thermo Fisher Scientific, Waltham, MA, USA). A panel of 13 cytokines was quantified with a bead-based ELISA kit (BioLegend, San Diego, CA, USA) following the manufacturer's protocol. Cytokine concentrations (pg/ml) were normalized to the corresponding sample total protein concentration (mg protein/mg tissue).

## RAW 264.7 macrophage culture

RAW 264.7 murine macrophages were cultured in a humidified incubator at 37°C and 5% $CO_2$ using DMEM supplemented with 10% heat-inactivated FBS, penicillin (200 U/ml), and streptomycin (200 µg/ml). Cells were passaged every 2–3 days and used within 10 passages after thawing. For the two-hit model experiment, RAW 264.7 cells were seeded into 24-well plates at a density of $2×10^5$ cells/ml and then treated with 1 mM I3A, followed by palmitate and LPS as reported before (*Krishnan et al., 2018*). For the p-AMPK activation experiment, RAW 264.7 cells were treated with 1 mM AICAR for 4 hr, followed by addition of 300 µM palmitate complexed with BSA. Following an 18 hr incubation, the cells were treated with 10 ng/ml LPS for another 6 hr.

## RNA extraction and qRT-PCR

Total RNA was extracted from RAW 264.7 cell pellets using the EZNA Total RNA Kit (Omega Bio-Tek, Norcross, GA, USA). Purity of isolated RNA was confirmed by A260/280 ratio. qRT-PCR analysis was carried out using the qScript One-Step PCR Kit (Quanta Biosciences, Gaithersburg, MD, USA) on a LightCycler 96 System (Roche, Indianapolis, IN, USA). Fold-change values were calculated using the

$2^{-\Delta\Delta Ct}$ method, with β-actin as the housekeeping gene. The primer sequences are listed in **Supplementary file 7**.

## AMPK western blot analysis

Cell pellets or lyophilized liver samples were lysed with modified RIPA buffer (50 mM Tris-HCl, pH 7.4, 1% Triton X-100, 150 mM NaCl, 1 mM EDTA, and 0.5% sodium deoxycholate) supplemented with a protease inhibitor cocktail (Sigma), 10 mM NaF, and 1 mM $Na_3VO_4$. The protein concentration was determined using the BCA Protein Assay Kit (Pierce). Protein samples were denatured with SDS and ~10 µg of protein was separated on a 10% SDS-PAGE gel. Proteins were transferred to a PVDF membrane (Thermo Fisher Scientific, Waltham, MA, USA) by wet-transfer electrophoresis. Non-fat milk (5%) in TBST solution was used to block non-specific binding. The blots were probed with appropriate primary antibodies (p-AMPK: 2535, total AMPK: 2603, β-actin: 12620, Cell Signaling Technology) and secondary antibody (anti-rabbit horseradish peroxidase-conjugated secondary antibody, 7074, Cell Signaling Technology). Proteins bound by both primary and secondary antibodies were visualized by chemiluminescence after incubating the blot with Clarity Max Western ECL Blotting Substrate (Bio-Rad, Hercules, CA, USA). Blot images were acquired on a ChemiDoc gel imaging system (Bio-Rad, Hercules, CA, USA). Proteins were quantified by normalizing the intensity of the protein band of interest to the intensity of the β-actin band in the same lane using the Image Lab software (Bio-Rad, Hercules, CA, USA).

## Small interfering RNA transfection

Raw 264.7 cells were seeded into 6-well or 24-well plates at ~30% confluence and cultured for 24 hr prior to transfection. Cells were transfected with ON-TARGETplus mouse prkaa1 siRNA (Dharmacon, Lafayette, CO, USA) or ON-TARGETplus non-targeting pool (negative control, Dharmacon, Lafayette, CO, USA) using the GenMute siRNA transfection reagent (SignaGen Laboratories, Rockville, MD, USA) according to the manufacturer's instruction. After 24 hr, the medium was replaced with siRNA-free growth medium and incubated for an additional 24–72 hr. The transfection efficiency was determined by monitoring the AMPK mRNA and protein levels using qRT-PCR and western blot, respectively.

## Statistical analysis

Determination of significant difference in the level of a metabolite between two experimental groups used the Student's t-test with FDR correction. KEGG pathway enrichment analysis of metabolomics data was performed using MetaboAnalyst 5.0 (*Pang et al., 2022*). PCA and PLS-DA of the metabolomics and proteomics data were conducted using the mixOmics R package (v6.10.9). Ellipses drawn to represent 95% confidence regions assumed Gaussian distribution of latent variables (for PLS-DA) or scores (for PCA). Significance of separation between treatment groups was determined by calculating a standardized Euclidean distance matrix on the coordinates of latent variables or scores and performing a pairwise PERMANOVA with 999 permutations on the distance matrix (*Anderson, 2017*) and Hotelling's $T^2$ tests. A p-value of 0.05 was set as the significance threshold for all statistical comparisons. Heatmaps for liver metabolome (only significantly changed features), fecal metabolome, and liver proteome were generated with auto-scaled data. The features were clustered using the k-means method with Pearson correlation as the similarity metric. For protein analysis, a VIP score was calculated for each protein based on the PLS-DA result, and proteins with a VIP score <1.2 were excluded from the clustering analysis to avoid overfitting. KEGG pathway enrichment analysis of proteomics data was performed using clusterProfiler R package (v3.14.3).

## Acknowledgements

We gratefully acknowledge the use of facilities at the Microbial Analysis, Resources, and Services (MARS) Center for Open Research Resources and Equipment at the University of Connecticut, Texas A&M High Performance Research Computing, Integrated Metabolomics Analysis Core (IMAC) at Texas A&M University, and Mass Spectrometry Core Lab at Tufts University.

## Additional information

### Funding

| Funder | Grant reference number | Author |
|---|---|---|
| Texas A and M University | Ray B. Nesbitt Endowed Chair | Arul Jayaraman |
| Tufts University | Karol Family Professorship | Kyongbum Lee |

The funders had no role in study design, data collection and interpretation, or the decision to submit the work for publication.

### Author contributions

Yufang Ding, Conceptualization, Formal analysis, Investigation, Methodology, Writing – original draft, Writing – review and editing; Karin Yanagi, Conceptualization, Formal analysis, Investigation, Writing – original draft, Writing – review and editing; Fang Yang, Formal analysis, Writing – original draft; Evelyn Callaway, Clint Cheng, Investigation, Methodology; Martha E Hensel, Formal analysis; Rani Menon, Robert C Alaniz, Formal analysis, Investigation; Kyongbum Lee, Arul Jayaraman, Conceptualization, Supervision, Visualization, Writing – original draft, Project administration, Writing – review and editing

### Author ORCIDs

Yufang Ding ⓘ http://orcid.org/0000-0001-6633-8192
Arul Jayaraman ⓘ http://orcid.org/0000-0001-9276-8284

### Ethics

All procedures were performed in accordance with Texas A&M University Health Sciences Center Institutional Animal Care and Use Committee guidelines under an approved animal use protocol (AUP #2017-0145).

Joint Public Review: https://doi.org/10.7554/eLife.87458.3.sa1
Author Response https://doi.org/10.7554/eLife.87458.3.sa2

## Additional files

### Supplementary files

• Supplementary file 1. Chromatography gradient method for untargeted LC-MS metabolomics. Solvent A was formic acid solution in water (0.1% vol/vol). Solvent B was a 0.1% vol/vol formic acid solution in methanol. The flow rate used was 0.4 ml/min.

• Supplementary file 2. Chromatography gradient and LC-MS parameters used for detection of bile acids. For the chromatographic method, solvent A was ammonium acetate (2 mM) in water and solvent B was methanol:acetonitrile (50:50, vol/vol). The flow rate used was 0.4 ml/min.

• Supplementary file 3. Chromatography gradient and LC-MS parameters for free fatty acid analysis. For the chromatographic method, solvent A was acetonitrile/water (3:2, vol/vol) containing 10 mM ammonium acetate. Solvent B was acetonitrile/isopropanol (1:1, vol/vol). The injection volume was 5 μl and the oven temperature was set to 55°C.

• Supplementary file 4. Chromatography gradient method used for untargeted proteomics analysis of liver tissue.

• Supplementary file 5. Data processing workflow used in untargeted proteomic analysis of liver tissue.

• Supplementary file 6. LC-MS parameters used for targeted proteomic analysis of liver tissue.

• Supplementary file 7. Primer sequences used for RT-PCR analysis of gene expression.

• MDAR checklist

## Data availability

Sequencing data has been deposited in the NCBI BioProject database (PRJNA1029996). Mass spectrometry proteomics data have been deposited to the ProteomeXchange Consortium via the PRIDE partner repository with the dataset identifier PXD046255.

The following datasets were generated:

| Author(s) | Year | Dataset title | Dataset URL | Database and Identifier |
|---|---|---|---|---|
| Ding Y | 2023 | Oral supplementation of indole-3-acetate alleviate diet induced steatosis and hepatic inflammation in mice | https://www.ncbi.nlm.nih.gov/bioproject/PRJNA1029996 | NCBI BioProject, PRJNA1029996 |
| Yanagi K, Lee K | 2024 | Liver proteiomic analysis on westeren diet fed mice under oral supplementation of indole-3-acetate | https://www.ebi.ac.uk/pride/archive/projects/PXD046255 | PRIDE, PXD046255 |

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
