## [Editor Report · eLife assessment]

The studies are **important** to the field of hepatic steatosis and inflammation. The data provided are **convincing** that treatment with I3A mitigated Western diet (WD)-induced hepatic steatosis, inflammation and reversed WD-induced alterations in liver bile acids and free fatty acids in mice.

---

## [Referee Report · Joint Public Review]

The study by Ding et al demonstrated that microbial metabolite I3A reduced western diet induced steatosis and inflammation in mice. They showed that I3A mediates its anti-inflammatory activities through AMP-activated protein kinase (AMPK)-dependent manner in macrophages. Translationally, they proposed that I3A could be a potential therapeutic molecule in preventing the progression of steatosis to NASH.

Major strengths

• Authors clearly demonstrated that the Western Diet (WD)-induced steatosis and I3A treatment reduced steatosis and inflammation in pre-clinical models. Data clearly supports these statements.

• I3A treatment rescued WD-altered bile acids as well as partially rescued the metabolome, proteome in the liver.

• I3A treatment reduced the levels of enzymes in fatty acid transport, de novo lipogenesis and β-oxidation

• I3A mediates its anti-inflammatory activities through AMP-activated protein kinase (AMPK)-dependent manner in macrophages.

Minor

I agree with the authors that investigating known other AhR ligands in comparison may be beyond the scope of this study.

---

## [Author Response]

The following is the authors’ response to the original reviews.

**Reviewer #1:**
1. Utilization of known AhR ligands as controls will strengthen the interpretation of the conclusions.

We agree with the reviewer that AhR ligands could be used as controls for delineating structure-activity relationships and cell context-specific effects. However, such studies are beyond the scope of the current manuscript. The AhR has many endogenous ligands, including several tryptophan derived metabolites, that have been shown to elicit different responses depending on the dose and cell type. Our unpublished data show that the expression of AhR target genes such as Cyp1a1, Cyyp2e1, and Tiparp were not modulated by I3A in RAW cells, which suggests that the observed effects may occur independent of the AhR.

**Reviewer #2:**
Specific comments:1. The title is misleading "Microbially-derived indole-3-actate" suggests that this article is about the production of I3A by the gut microbiota, in fact this is a dietary supplementation article. The title needs to reflect this fact.

Our title reflects the natural source of I3A in mice. We used oral supplementation to study the effects of this metabolite. Per suggestion by the reviewer, we changed the title as follows:

“Oral supplementation of gut microbial metabolite indole-3-acetate alleviates diet-induced steatosis and inflammation in mice”

2). The description of the amount of I3A in the drinking water is not properly described. The actual concentration in the drinking water should be given.

The concentration of I3A in drinking water was as follows: WD50 = 0.5mg/ml and WD100 = 1mg/ml. We added this information in the revised manuscript.

1. The serum concentration data of I3A is critical data and should be moved in Figure 1.

We have now included serum levels of I3A as part of Figure 1.

1. The authors should have determined the actual concentration of indole-3-actetate in serum by running a standard curve of I3A during the LC-MS analysis. Also, recovery and matrix effects should be determined. Without this information their data will be difficult to compare to other studies.

We agree with the reviewer that quantification of I3A in serum would be useful. However, we are unable to do so due to limited sample available as well as concerns with sample integrity after long-term storage.

1. In the data in Figure S1C, there appears to be only 2-3 mice out of nine that exhibit a difference in serum indole-3-acetate levels between the WD-50 and WD-100. Do the authors have an explanation for this small difference compared to the other endpoints assessed?

The serum I3A measurements at week 16 are a snapshot that may not reflect tissue levels due to differences in water intake, I3A metabolism in the body, and/or elimination of I3A. The other phenotypic assays are physiological measurements that reflect the result of sustained administration of I3A.

1. Since the Ah receptor may play a role in the results obtained CYP1A1 mRNA levels in the liver and intestinal tract should have been measured.

We measured alterations in Cyp1a1 mRNA in the liver and no significant change was observed in the WD50 and WD100 groups relative to controls. Also, see response to reviewer 1.

1. The main mechanistic experiment performed is shown in Figure 6 and the figure legend states that they are examining macrophages, but these are cell lines, they are macrophages models, and this should be clearly stated. The first two panels are liver data, so the title of the figure legend needs to reflect that fact.

We agree and have changed the title of Figure 6 to “I3A modulates AMPK phosphorylation and suppresses RAW 264.7 macrophage cell inflammation in an AMPK dependent manner”.

1. In Figure 6, 1 mM I3A is added to the cells, how is this very high concentration relevant to the concentrations observed in vivo? Does adding 1 mM acetate to the cell culture media lower the pH of the media and could this influence the results obtained? Would acetic acid yield the same results? Could treatment with an acid even explain in vivo results?

It is difficult to match the concentration of I3A in the in vitro experiments to liver tissue concentrations. Addition of 1 mM I3A did not lower the pH of cell culture media or reduce the viability of cultured RAW 264.7 macrophages. As I3A is not known to degrade into acetic acid and indole, we do not expect acetic acid to recapitulate the effects elicited by I3A.

**Reviewer #3:**
My primary concern with the manuscript is the organization and interpretation of the data. It appears that little effort was given by the authors on interpreting the data and digesting it for the reader into a coherent package. Rather, the authors have collected a vast amount of data and organized it without much thought about what the reader would take away from it. Furthermore, it seems the authors have taken this as an opportunity to overload this manuscript with data that are superfluous to the conclusions the authors draw at the end. Based on this, I think the authors need to invest more time into distilled their complex biological data into a unifying scientific interpretation for the readers that advances our understanding of I3A. My suggestions for the authors are described below.1. The data lack a rationale behind how they are organized within the manuscript. For example, the authors will combine disparate biological pathways and lump data together without logic as in Figure 2. Why are inflammatory pathways and bile acid synthesis combined in a figure? What was the rationale?

We respectfully disagree that the data are presented without rationale. Both inflammation and bile acid dysregulation are commonly observed with NAFLD and thus are presented in two separate panels of Figure 2 (A, inflammatory cytokines, and B bile acids).

1. The authors give very little effort to performing integrative omics analysis even though multi-omics is provided. Example given, the authors provide proteomic data on the fatty acid metabolism pathway, however, no mention of this pathway within the metabolomic dataset. Vice versa, the authors provide in depth investigation in the metabolic changes within the tryptophan pathway, however, no investigation into the proteomic changes that may underlie this phenomenon. It would be recommended that the authors invest more energy into performing more in-depth analysis of their multi-omics data presented.

We attempted to co-analyze the proteomic and metabolomic data, but this analysis was not informative. Protein and metabolite abundances do not necessarily correlate, and the two types of omics data carry different observation biases. For example, label-free, untargeted proteomics data favor abundant proteins, whereas untargeted metabolomics data are influenced by concentration and ionization efficiency, among other factors. Therefore, we opted to analyze the two datasets independently, and then linked the findings from the two analyses using biological pathways as guides. For example, we describe changes in acyl-carnitine and discuss how this observation is consistent with changes in abundance of fatty acid metabolism enzymes.

1. Figures 1&2 shows that low dose treatment reduces inflammation but does not alter hepatic TG levels. This is in direct disagreement with the graphical model provided by the authors (Supp. Fig 9). In the author's model, I3A is directing hepatic lipid metabolism through modulation of macrophage inflammation. This interpretation is erroneous and needs to be reevaluated by the authors. Furthermore, the tryptophan pathway and bile acid pathways are not even represented in the model, which begs the question of why that data are included in the manuscript to begin with.

We would like to respectfully point out that Figure 1D does show a statistically significant (p < 0.05) difference in liver TG between the WD and WD100 groups. Supp. Figure S9 is meant to be a summary of the main biochemical changes elicited by I3A that we have shown in the current study (e.g., the involvement of AMPK) rather an atlas of all the changes detected in the metabolomics and proteomic data. Specifically, we have not included the tryptophan or bile acid pathways as we do not have mechanistic information on how these changes are mediated by I3A.

1. The authors switch from hepatocytes to macrophages without giving any rationale, The authors need to invest more time into describing a logical flow of thought when assembling the manuscript.

We mention the rationale for investigating the effect of I3A on macrophages in the introduction (last paragraph of the section): “In vitro, both I3A and TA attenuated the expression of inflammatory cytokines (Tnfα, Il-1β and Mcp-1) in macrophages exposed to palmitate and LPS.”. We also explain why we used an in vitro model, RAW cells, at the beginning of the corresponding Results section: “Since our previous study found that the metabolic effects of I3A in hepatocytes depend on the AhR, we tested if this was also the case in macrophages.” Moreover, the strong effects of I3A on liver inflammatory cytokines also motivates the macrophage experiments.